# Global Dynamics of SARS-CoV-2 Infection with Antibody Response and the Impact of Impulsive Drug Therapy

**DOI:** 10.3390/vaccines10111846

**Published:** 2022-10-31

**Authors:** Amar Nath Chatterjee, Fahad Al Basir, Dibyendu Biswas, Teklebirhan Abraha

**Affiliations:** 1Department of Mathematics, K.L.S. College, Nawada, Magadh University, Bodhgaya 805110, Bihar, India; 2Department of Mathematics, Asansol Girls’ College, Asansol 713304, West Bengal, India; 3Department of Mathematics, City College of Commerce and Business Administration, 13, Surya Sen Street, Kolkata 700012, West Bengal, India; 4Department of Mathematics, Addis Ababa Science and Technology University, Addis Ababa P.O. Box 16417, Ethiopia; 5Department of Mathematics, Aksum University, Aksum P.O. Box 1010, Ethiopia

**Keywords:** epithelial cell, antibody response, basic reproduction number, transcritical bifurcation, impulsive control, drug holidays

## Abstract

Mathematical modeling is crucial to investigating tthe ongoing coronavirus disease 2019 (COVID-19) pandemic. The primary target area of the SARS-CoV-2 virus is epithelial cells in the human lower respiratory tract. During this viral infection, infected cells can activate innate and adaptive immune responses to viral infection. Immune response in COVID-19 infection can lead to longer recovery time and more severe secondary complications. We formulate a micro-level mathematical model by incorporating a saturation term for SARS-CoV-2-infected epithelial cell loss reliant on infected cell levels. Forward and backward bifurcation between disease-free and endemic equilibrium points have been analyzed. Global stability of both disease-free and endemic equilibrium is provided. We have seen that the disease-free equilibrium is globally stable for $R_{0}<1$, and endemic equilibrium exists and is globally stable for $R_{0}>1$. Impulsive application of drug dosing has been applied for the treatment of COVID-19 patients. Additionally, the dynamics of the impulsive system are discussed when a patient takes drug holidays. Numerical simulations support the analytical findings and the dynamical regimes in the systems.

## 1. Introduction

COVID-19 is considered to be transmitted mainly between people who are close in contact with one another within about six feet, as well as through respiratory droplets created when an infected person coughs or sneezes. These droplets can enter the mouths or noses of people close by or possibly be inhaled into the lungs [1]. Virus spread depends on the possibility of touching virus-infected surfaces or objects and then touching one’s own mouth, nose, or possibly eyes [2,3].

In a SARS-CoV-2-infected human, the innate and adaptive immune responses work together to neutralize the threat of SARS-CoV-2 infection [4,5,6]. When the virus enters the human body, the innate immune response starts immediately. Proteins of the natural immune system in a healthy cell also respond against the invading pathogens within the first minutes or hours of infection [7]. This response is of great importance in preventing new infections through the activation of the adaptive immune system [8,9]. Cytokines, which are small soluble proteins, are an essential component of the immune system [10]. They are secreted from different cells in the human body. They can be categorized into one of four families: (i) the hematopoietic family, (ii) the immunoglobin superfamily, (iii) the tumor necrosis factor family, and (iv) interferons (IFNs) [11]. Cytokines balance the innate and adaptive immune responses. Among cytokines, IFNs play a vital role in the innate immune response during viral infection. Thus, we consider the effect of adaptive immune responses in our mathematical model.

There are many research articles that include population modeling for the transmission dynamics of COVID-19 [12,13,14,15,16,17,18] These studies mainly focused on susceptible exposed populations and asymptomatic infected populations for a particular region. Population density is a major factor for disease transmission for this type of modeling. Some of these articles include vaccinations and optimal control [19,20,21]; additionally, some articles focus on population awareness through media [22,23].

However, for the SARS-CoV-2 dynamics at the micro-level (i.e., the dynamics of the disease within the human host), few model-based studies are available. The dynamics of SARS-CoV-2 infection can give insight to controlling the virus in a human host [11,24,25]. Many infectious disease dynamics are explored extensively by researchers with the help of mathematical modeling with real data at the cellular level [26,27,28,29]. Tang et al. [30] proposed a four-population host cell infection model for MERS-CoV mediated by DPP4 receptors. The infection processes of SARS-CoV-2, SARS-CoV, and MERS-CoV are similar. Researchers are still working on inter-host modeling for SARS-CoV-2 and target cell limitations under immune responses [31,32]. Hernandez et al. [31] proposed a model to examine cellular level dynamics and T cell responses against viral replication. Wang et al. [32] evaluated the effect of several potential interventions for SARS-CoV-2. Their study reveals that combining antiviral drugs with interferons effectively reduces the viral plateau phase and shortens the recovery time. Chatterjee and Bashir [29] formulated a mathematical model to examine the consequences of adaptive immune response to viral mutation to control disease transmission. They also studied the effect of antiviral drug therapy and its impact on model dynamics. Chatterjee et al. [11] proposed a set of fractional differential equation models considering uninfected epithelial cells, infected epithelial cells, SARS-CoV-2 virus, and CTL response cells, accounting for the lytic and non-lytic effects of immune responses [11]. They also studied the impact of a commonly used antiviral drug in COVID-19 treatment in an optimal control-theoretic approach. Wang et al. [28] studied the effect of antiviral drugs against SARS-CoV-2 viral dynamics during COVID-19 infection. In [33], within-host dynamics of SARS-CoV-2 infection were studied with potential treatments. Authors have used repurposed drugs (remdesivir) that inhibit the transcription of SARS-CoV-2.

Despite numerous therapeutic strategies, to date, there is no specific effective treatment for SARS-CoV-2 infection. Recently, all over the world, clinicians have been working on an effective therapy for coronavirus disease 2019 (COVID-19). Clinical observation suggests that cytokine levels enhance the hyperinflammatory response secondary to SARS-CoV-2 infection. This is the leading cause of multi-organ damage for COVID-19 patients [1]. For these reasons, numerous clinical trials are currently undergoing to explore the effectiveness of drugs such as interleukin-1 blockers and interleukin-6 inhibitors in COVID-19 [33,34,35].

The most useful method to study drug dynamics is the use of impulsive differential equations [28]. Perfect or imperfect drug adherence and drug holidays can make the development of resistance easy. Recently, the effects of perfect drug dosing on antiretroviral therapy have been studied by impulsive differential equations. Chatterjee and Basir [24] formulated a dynamic model of epithelial cells during SARS-CoV-2 infection and CTL responses. They established a new mathematical model considering epithelial cells and the role of the ACE2 receptor using impulsive differential equations, which describe the within-host dynamics of SARS-CoV-2 infection with treatments. The dosing period and threshold values of dosage can be obtained using this method.

In the present research, we study the dynamics of COVID-19 in humans using immunostimulant drugs. We explor the dynamics using the antibody-response model of SARS-CoV-2 infection by examining the interaction between viral replication. We consider uninfected target epithelial cells, infected epithelial cells, SARS-CoV-2 virus, and antibody responses in the modeling process, with an aim to reduce the infected epithelial cells and viral load using immunostimulant drugs. The local and global dynamics of the system without drugs have been provided. Forward transcritical bifurcation is also analyzed. Finally, we implement impulsive differential equations to observe the impact of drugs. The dosing rate and interval, and how many drug holidays a patient takes have been studied.

The drug data we have used here are those of monoclonal antibodies (mAbs). These have the capability to detect and prevent the disease propagation [36,37]. A SARS-CoV-2 patient who is at high risk of transmitting to another individual with SARS-CoV-2 for longer than 4 weeks, and who is unable to mount an adequate immune response to SARS-CoV-2 vaccination, can take an initial dose of 600 mg of casirivimab and 600 mg of imdevimab, then repeat doses of 300 mg of casirivimab and 300 mg of imdevimab once every 4 weeks [38].

The article is organized as follows. The next section (Section 2) contains the formulation of a mathematical model of immune responses. The qualitative properties of the model are provided in Section 3. Theoretical analysis of the impulsive model is carried out in Section 4. The numerical simulation is included based on the analytical findings in Section 5. Discussion and concluding remarks are given in Section 6.

## 2. Derivation of the Mathematical Model

The mathematical model helps us to understand the basic dynamics of viral infection. In general cases, modeling consist of a antibody response model with some variants [39]. Here, we consider the simplest version including three populations, namely:$E_{S}\left(t\right)$: the uninfected susceptible target cells, which are surface epithelial cells with ACE-2 receptors located in the respiratory tract, including the lungs and nasal and trachea/ bronchial tissues;$E_{I}\left(t\right)$: the SARS-CoV-2-infected virus-producing cells;V(t): the virus particles.

The SARS-CoV-2 dynamics with immune cells are proposed in [39] as the following
(1)$$\begin{array}{ccc} \frac{dE_{S}}{dt} & = & \Pi -\beta E_{S}V-\mu_{1}E_{S};\;\\ \frac{dE_{I}}{dt} & = & \beta E_{S}V-\mu_{2}E_{I};\;\;\\ \frac{dV}{dt} & = & pE_{I}-\mu_{3}V. \end{array}$$

The first equation of (Equation 1) shows the dynamics of uninfected epithelial cells ($E_{S}\left(t\right)$); the second equation shows the dynamics of the infected epithelial cells ($E_{I}\left(t\right)$). The replication of the SARS-CoV-2 virus (V(t)) in the third equation of (Equation 1) is considering, as SARS infection promotes endothelins on several organs as a direct consequence of viral involvement [31].

The growth rate of epithelial cell is denoted as Π. The virus infects the uninfected cells with a rate β ($mL\;{\left(RNA\;copies\right)}^{-1}\;day^{-1}$). After a cell becomes infected, it behaves as a virus-producing cell and produces viruses at a rate *p* ($day^{-1}$), and are virus particle cleared at a rate $\mu_{3}$ (day${}^{-1}$). The uninfected susceptible cells are cleared at a rate $\mu_{1}$ (day${}^{-1}$) due to their natural apoptosis, and the infected cells are removed from the system at a rate $\mu_{2}$ (day${}^{-1}$) as a result of cytopathic viral effects and immune response [31].

Cytokine is vital in inhibiting viral replication and modulating downstream effects of the immune response. Specific cytokines activate natural killer cells (NK), which act against virally infected cells. In the case of SARS-CoV-2 infection, it is observed that viruses often target the JaK/STAT pathway (i.e., a chain of interactions between proteins in a cell) to decrease the production of IFNs. This immune suppressing mechanism observed in SARS-CoV-2 can be represented in the functional form of a decrease in the cytokine production rate, assumed to be $\frac{\alpha E_{I}}{V+\theta }$. Cytokines activate the adaptive immune system, mainly T cells and B lymphocytes, to produce an antibody that acts against the virus. B cells mainly secrete IgM and IgG antibodies that are released from blood and lymph fluid and neutralize the SARS-CoV-2 viral particles.

Considering the antibody responses A(t), we extend the antibody-response model to include the depletion of viruses modeled via the term rAV. The extended model of (Equation 1) reads as follows:(2)$$\begin{array}{c} \left\{\begin{array}{ccc} \frac{dE_{S}}{dt} & = & \Pi -\beta E_{S}V-\mu_{1}E_{S},\\ \frac{dE_{I}}{dt} & = & \beta E_{S}V-\mu_{2}E_{I},\;\;\;\;\;\;\;\;\\ \frac{dV}{dt} & = & pE_{I}-\mu_{3}V-rVA,\\ \frac{dA}{dt} & = & \frac{\alpha E_{I}}{V+\theta }-\mu_{4}A,\;\;\;\;\;\;\;\;\; \end{array}\right. \end{array}$$
with the initial condition
(3)$$\begin{array}{ccc} E_{S}\left(0\right) & = & E_{S0},\;E_{I}\left(0\right)=E_{I0},\;V\left(0\right)=V_{0},\;A\left(0\right)=A_{0}. \end{array}$$

The graphical representation of the above model is shown in Figure 1.

Here, *r* is the rate at which the antibody neutralizes the viral particles, α is the antibody simulation rate constant, and θ is the strength of antibody suppression, in which the antibodies are lost at a rate of $\mu_{4}$.

We now impose impulsive drug dosing in the above model and analyze it in Section 4.

Impulsive differential equations result from drug effects; the metabolites are assumed to decay with time in an exponential manner during each cycle and are assumed to change instantaneously at dosing times $t_{j}$ for different drug doses, which can result in either implicit or explicit models. In the presence of antibody-controlled therapy with perfect adherence, we consider Model (Equation 2). Before analyzing the system, we first discuss the one-dimensional impulse system as follows:(4)$$\begin{array}{ccc} \frac{dE_{S}}{dt} & = & \Pi -\beta E_{S}V-\mu_{1}E_{S},\;\;\;\;t\neq t_{n}\\ \frac{dE_{I}}{dt} & = & \beta E_{S}V-\mu_{2}E_{I},\;\;\;\;\;\;\;t\neq t_{n}\\ \frac{dV}{dt} & = & pE_{I}-\mu_{3}V-rVA,\;\;\;\;t\neq t_{n}\\ \frac{dA}{dt} & = & \frac{\alpha E_{I}}{V+\theta }-\mu_{4}A,\;\;\;\;\;\;\;\;t\neq t_{n},\\ \frac{dA}{dt} & = & \zeta D+\frac{\alpha E_{I}}{V+\theta }-\mu_{4}A,\;\;\;\;t=t_{n},\\ \frac{dD}{dt} & = & -\mu_{5}D,\;\;\;\;\;\;\;\;\;\;\;\;\;\;\;\;\;\;\;t\neq t_{n}\\ D(t_{n}^{+}) & = & \omega +D\left(t_{n}^{-}\right),\;\;\;\;\;\;\;\;\;\;\;\;\;\;\;\;\;\;\;t=t_{n}. \end{array}$$
$D(t_{n}^{-})$ denotes the drug concentration immediately before the impulse drug dosing, $D(t_{n}^{+})$ denotes the concentration after the impulses, and ω is the dose that is taken at each impulse time $t_{n}$, n∈N. Here, ζ is the rate at which antibodies are produced due to the use of a drug.

## 3. Dynamics of the Model without Impulses

In this section, basic properties such as nonnegativity and boundedness, basic reproduction number, and equilibrium points and their stability properties are analyzed.

### 3.1. Non-Negativity and Boundedness

In this section we investigate the non-negativeness of the state variables of the Model (Equation 2) for all time *t* with initial condition $(E_{S}\left(0\right),E_{I}\left(0\right),V\left(0\right),A\left(0\right)\in R_{+}^{4}$. To prove the non-negativity property, we establish the following theorem.

**Theorem** **1.***Model* (Equation 2) *with initial condition* (Equation 3) *satisfies*
$E_{S}\left(t\right)\geq 0,\;E_{I}\left(t\right)\geq 0,\;V\left(t\right)\geq 0,\;A\left(0\right)\geq 0$ for all t>0*; then, Model* (Equation 2) *is positively invariant*.

**Proof.** The first equation of Model (Equation 2) can be rewritten as
(5)$$\begin{array}{ccc} \frac{dE_{S}\left(t\right)}{dt} & = & \Pi -\beta E_{S}\left(t\right)V\left(t\right)-\mu_{1}E_{S}\left(t\right),\\ & = & \Pi -\xi_{1}E_{S}\left(t\right), \end{array}$$
where $\xi_{1}=\beta V\left(t\right)+\mu_{1}$. Integrating (Equation 5), we obtain
(6)$$\begin{array}{ccc} E_{S}\left(t\right) & = & E_{S}\left(0\right)\exp\left(-\int_{0}^{t}\xi_{1}\left(u\right)du\right)+\Pi \exp\left(-\int_{0}^{t}\xi_{1}\left(u\right)du\int_{0}^{t}\left(e^{\int_{0}^{t}\xi_{1}\left(v\right)dv}\right)du\right)>0. \end{array}$$This implies that $E_{S}\left(t\right)$ is nonnegative for all *t*. For the second equation of Model (Equation 2), we have
$$\begin{array}{ccc} \frac{dE_{I}\left(t\right)}{dt} & \geq & -\mu_{2}E_{I}\left(t\right), \end{array}$$
which gives
(7)$$\begin{array}{ccc} E_{I}\left(t\right) & \geq & E_{I}\left(0\right)\exp\left(-\int_{0}^{t}\mu_{2}du\right)>0. \end{array}$$The third equation of Model (Equation 2) can be written as
(8)$$\begin{array}{ccc} \frac{dV(t)}{dt} & = & pE_{I}\left(t\right)-\mu_{3}V\left(t\right)-rA\left(t\right)V\left(t\right),\\ & = & pE_{I}\left(t\right)-\xi_{2}V\left(t\right), \end{array}$$
where $\xi_{2}=\mu_{3}+rA\left(t\right)$. Integrating (Equation 8), we obtain
(9)$$\begin{array}{ccc} V(t) & = & V\left(0\right)\exp\left(-\int_{0}^{t}\xi_{2}\left(u\right)du\right)+pE_{I}\left(t\right)\exp\left(-\int_{0}^{t}\xi_{2}\left(u\right)du\int_{0}^{t}\left(e^{\int_{0}^{t}\xi_{2}\left(v\right)dv}\right)du\right)>0. \end{array}$$This implies that V(t) is non-negative for all *t*. In a similar way, for the last equation, we can say that
$$\begin{array}{ccc} \frac{dA(t)}{dt} & \geq & -\mu_{4}A\left(t\right), \end{array}$$
which gives
(10)$$\begin{array}{ccc} A(t) & \geq & A\left(0\right)\exp\left(-\int_{0}^{t}\mu_{4}du\right)>0. \end{array}$$The above results show that all the solution trajectories of Model (Equation 2) are non-negative for all t>0. □

To verify the boundedness of Model (Equation 2) with non-negative initial values, we use the following theorem.

**Theorem** **2.***Model* (Equation 2) *with the initial condition* (Equation 3) *is uniformly bounded in the positive invariant set*
U*, where*
(11)$$\begin{array}{ccc} U & = & \left\{\left(E_{S}\left(t\right),E_{I}\left(t\right),V\left(t\right),A\left(t\right)\right)\in R_{+}^{4}|0\leq E\leq \frac{\Pi }{\mu_{1}},0\leq V\left(t\right)\leq \frac{p\Pi }{\mu_{1}\mu_{3}},0\leq A\left(t\right)\leq \frac{\alpha \Pi }{\mu_{1}\mu_{4}\theta }\right\}. \end{array}$$

**Proof.** From the positivity of the solution, we obtain
(12)$$\begin{array}{ccc} \frac{dE_{S}\left(t\right)}{dt} & = & \Pi -\mu_{1}E_{S}\left(t\right)-\beta E_{S}\left(t\right)V\left(t\right),\\ & \leq & \Pi -\mu_{1}E_{S}. \end{array}$$This implies that
(13)$$\begin{array}{ccc} \lim_{t\to \infty }\sup E_{S}\left(t\right) & \leq & \frac{\Pi }{\mu_{1}}. \end{array}$$Now, $E\left(t\right)=E_{S}\left(t\right)+E_{I}\left(t\right)$; then,
(14)$$\begin{array}{ccc} \frac{dE(t)}{dt} & = & \Pi -\mu_{1}E_{S}\left(t\right)-\mu_{2}E_{I}\left(t\right),\\ & \leq & \Pi -\mu \left(E_{S}\left(t\right)+E_{I}\left(t\right)\right)\;\;\mathrm{where},\mu =\min\{\mu_{1},\mu_{2}\},\\ & \leq & \Pi -\mu E(t). \end{array}$$Hence, we can write $\lim_{t\to \infty }\sup E\left(t\right)\leq \frac{\Pi }{\mu_{1}}$.From the third equation of (Equation 2), we also have
(15)$$\begin{array}{ccc} \frac{dV(t)}{dt} & = & pE_{I}\left(t\right)-\left(\mu_{3}+rA\left(t\right)\right)V\left(t\right),\\ & \leq & pE_{I}\left(t\right)-\mu_{3}V\left(t\right),\\ \Rightarrow \frac{dV(t)}{dt}+\mu_{3}V\left(t\right) & \leq & pE_{I}\left(t\right)\\ \Rightarrow \frac{dV(t)}{dt}+\mu_{3}V\left(t\right) & \leq p\frac{\Pi }{\mu_{1}}\\ \Rightarrow \lim_{t\to \infty }\sup V\left(t\right) & \leq & \frac{p\Pi }{\mu_{1}\mu_{3}}. \end{array}$$From the last equation of Model (Equation 2), we obtain
$$\begin{array}{cc} \frac{dA(t)}{dt}= & \frac{\alpha E_{I}}{V+\theta }-\mu_{4}\;A\\ \leq & \frac{\alpha E_{I}}{\theta }-\mu_{4}A \end{array}$$This implies that
$$\begin{array}{cc} \frac{dA}{dt}+\mu_{4}A\leq & \frac{\alpha E_{I}}{\theta }\leq \frac{\alpha }{\theta }\left(\frac{\Pi }{\mu_{1}}\right). \end{array}$$Hence,
$$\lim_{t\to \infty }\sup A\left(t\right)\leq \frac{\alpha \Pi }{\mu_{1}\mu_{4}\theta }.$$Therefore, all the solution trajectories that start from $R_{+}^{4}$ will enter the region U and never leave it.  □ 

### 3.2. Basic Reproduction Number

The basic reproductive number ($R_{0}$) is useful in estimating the ability of a new pathogen to be transmitted. It is defined as the average number of secondary transmissions from a single infected person. When $R_{0}$ is less than one, then the disease (epidemic) does not grow, but if it is greater than one, the disease grows. The basic reproductive number has important implications for disease control. It indicates the level of mitigation efforts needed to bring an epidemic under control [40].

The next-generation matrix method, introduced by Driessche, Pauline, and Watmough in [41], is used to determine the basic reproduction number. For this purpose, we consider the non-negative matrix G and non-negative *M* matrix H, which represents the production of the new infection and its transportation, respectively. The viral dynamical system of (Equation 2) is defined below:(16)$$\begin{array}{ccc} G & = & \left(\begin{array}{c} \beta E_{S}V\\ 0 \end{array}\right),\;\;H=\left(\begin{array}{c} \mu_{2}E_{I}\\ -pE_{I}+\left(\mu_{3}+rA\right)V \end{array}\right). \end{array}$$

Now, the matrix G and H can be written as
(17)$$\begin{array}{ccc} G & = & \left[\frac{\partial G_{i}}{\partial x_{j}}\left(\bar{P}\right)\right],\;H=\left[\frac{\partial H_{i}}{\partial x_{j}}\left(\bar{P}\right)\right],\mathrm{with},\;1\leq i,j\leq 2. \end{array}$$

Additionally, G is non-negative and H is a non-singular *M* matrix; all eigenvalues of $J_{4}$ have positive real parts.

For our system,
(18)$$\begin{array}{ccc} G & = & {\left[\begin{array}{cc} 0 & \beta E_{S}\\ 0 & 0 \end{array}\right]}_{\bar{P}}=\left[\begin{array}{cc} 0 & \frac{\beta \Pi }{\mu_{1}}\\ 0 & 0 \end{array}\right], \end{array}$$
(19)$$\begin{array}{ccc} H & = & {\left[\begin{array}{cc} \mu_{2} & 0\\ -p & \mu_{3}+rA \end{array}\right]}_{\bar{P}}=\left[\begin{array}{cc} \mu_{2} & 0\\ -p & \mu_{3} \end{array}\right]. \end{array}$$

Therefore, the basic reproduction number, denoted by $R_{0}$, is the spectral radius of the next generation matrix and is obtained as
(20)$$\begin{array}{ccc} R_{0} & = & \rho \left(GH^{-1}\right)=\frac{p\beta \Pi }{\mu_{1}\mu_{2}\mu_{3}}. \end{array}$$

**Remark** **1.**
*Notice that the basic reproduction number $R_{0}$ is proportional to the infection rate β and replication rate p, and inversely proportional to the death rate of the virus $\mu_{3}$. Therefore, the disease can be managed by reducing the infection rate and replication rate p or increasing the death rate of the virus. This can be done using antiviral drugs. We adopted impulsive periodic application of antiviral drugs.*


### 3.3. Existence of Equilibrium Points

Model (Equation 2) has two equilibria: (i) the disease-free equilibrium $\bar{P}\left(\frac{\Pi }{\mu_{1}},0,0,0\right)$ and (ii) the endemic equilibrium point $P^{*}\left(E_{S}^{*},E_{I}^{*},V^{*},A^{*}\right)$, where
(21)$$\begin{array}{ccc} E_{S}^{*} & = & \frac{\Pi }{\beta V^{*}+\mu_{1}},\;E_{I}^{*}=\frac{\Pi \beta V^{*}}{\mu_{2}\left(\beta V^{*}+\mu_{1}\right)},A^{*}=\frac{\alpha \beta \Pi V^{*}}{\mu_{2}\mu_{4}\left(V^{*}+\theta \right)\left(\beta V^{*}+\mu_{1}\right)}, \end{array}$$
and $V^{*}$ satisfies the equation
(22)$$\begin{array}{ccc} b_{0}{V^{*}}^{2}+b_{1}V^{*}+b_{2} & = & 0, \end{array}$$
where
(23)$$\begin{array}{ccc} b_{0} & = & \mu_{2}\mu_{3}\beta ,\\ b_{1} & = & \mu_{2}\mu_{3}\left(\beta \;\theta +\mu_{1}\right)-\Pi \;\beta \;p+\frac{\Pi \;\alpha \;\beta \;r}{\mu_{4}}, \end{array}$$
(24)$$\begin{array}{ccc} b_{2} & = & \theta \;\left(-\Pi \;\beta \;p+\mu_{1}\mu_{2}\mu_{3}\right),\\ & = & \theta \mu_{1}\mu_{2}\mu_{3}\left(1-R_{0}\right). \end{array}$$

We have the following Theorem:

**Theorem** **3.**
*When $R_{0}>1$, one and only one endemic equilibrium $P^{*}$ exists. For $R_{0}<1$, there may exist two endemic points.*


**Proof.** From Equation (Equation 22), it is clear that $b_{0}>0$ and $b_{2}>0$ if $R_{0}<1$. Furthermore, if $R_{0}>1$, then $b_{2}<0$. Using Descartes’ rule of signs, we can say that there exist a unique endemic equilibrium if $b_{2}<0$ and two positive endemic equilibrium if $b_{2}>0,\;b_{1}<0$ and $b_{1}^{2}-4b_{0}b_{2}>0$. □

**Remark** **2.***Moreover, a transcritical bifurcation occurs when $b_{2}=0$; i.e., $R_{0}=1$ and $b_{1}<0$ with $b_{1}^{2}-4b_{0}b_{2}=0$ (the point where two positive endemic equilibrium coincide with each other and leave the stable disease-free equilibrium point)*.

### 3.4. Stability of Equilibrium Points

In this section, we discuss the local and global stability of the equilibrium points.

For the stability of disease-free equilibrium, we have the following theorem.

**Theorem** **4.**
*The disease-free equilibrium $\bar{P}\left(\frac{\Pi }{\mu_{1}},0,0,0\right)$ is locally asymptotically stable for $R_{0}<1$; when $R_{0}>1$, the disease-free system becomes unstable.*


**Proof.** To verify the local asymptotic stability at $\bar{P}$, we compute the Jacobian matrix of (Equation 2) around $\bar{P}$ as given below
(25)$$\begin{array}{ccc} J_{\bar{P}} & = & \left[\begin{array}{cccc} -\mu_{1} & 0 & -\frac{\beta \;\pi }{\mu_{1}} & 0\\ 0 & -\mu_{2} & \frac{\beta \;\pi }{\mu_{1}} & 0\\ 0 & p & -\mu_{3} & 0\\ 0 & \frac{\alpha }{\theta } & 0 & -\mu_{4} \end{array}\right]. \end{array}$$The characteristic equation from $\det(J_{\bar{P}}-\lambda I_{4})=0$ is
$$\left|\begin{array}{cccc} \lambda +\mu_{1} & 0 & \frac{\beta \;\pi }{\mu_{1}} & 0\\ 0 & \lambda +\mu_{2} & -\frac{\beta \;\pi }{\mu_{1}} & 0\\ 0 & -p & \lambda +\mu_{3} & 0\\ 0 & -\frac{\alpha }{\theta } & 0 & \lambda +\mu_{4} \end{array}\right|=0.$$
that is,
$$\begin{array}{c} \left(\lambda +\mu_{1}\right)\left(\lambda +\mu_{4}\right)\left(\lambda^{2}+\left(\mu_{2}+\mu_{3}\right)\lambda +\frac{1}{\mu_{1}}\left(\mu_{1}\mu_{2}\mu_{3}-\pi \;\beta \;p\right)\right) \end{array}=0,$$
with two eigenvalues $\lambda_{1}=-\mu_{1}<0$ and $\lambda_{2}=-\mu_{4}<0$; the rest of the spectrum is given by the roots of the transcendental equation
$$\lambda^{2}+a_{1}\lambda +a_{2}=0,$$
where $a_{1}=\mu_{2}+\mu_{3},\;a_{2}=\mu_{2}\mu_{3}\left(1-R_{0}\right)$. Here, $a_{1}=\mu_{2}+\mu_{3}>0$ and $a_{2}>0$ if $R_{0}<1$, which suggest that the two roots are negative real roots or have negative real parts. Hence, the disease-free equilibrium is locally asymptotically stable if $R_{0}<1$, and unstable if $R_{0}>1$. □

We have already proven that when $R_{0}<1$, the disease-free equilibrium $\bar{P}$ is locally asymptotically stable. Now, we verity the global stability of $\bar{P}$. For this purpose, we construct the Lyapunov function following [42,43]. We have the following theorem for the global stability of $\bar{P}$.

**Theorem** **5.**
*The disease-free equilibrium $\bar{P}$ is globally asymptotically stable if $R_{0}<1$ and it is a unique equilibrium. Otherwise, $\bar{P}$ is unstable and a unique endemic equilibrium $P^{*}$ exists.*


The proof of Theorem 5 is provided in Appendix A.

Now, we analyze the transcritical bifurcation between the disease-free and the endemic equilibrium points. We have the following theorem for this analysis.

**Theorem** **6.**
*The system exhibits forward transcritical bifurcation when $R_{0}=1.$*


**Proof.** To prove this theorem, we use the approach used by Castillo-Chavez and Song, 2004 [44] of applying the center manifold theory to analyze the dynamics of Model (Equation 2). The variables of Model (Equation 2) are transformed as $x_{1}=E_{S},\;x_{2}=E_{I},\;x_{3}=V,\;x_{4}=A$, and the total population $n=\sum_{i=1}^{4}x_{i}$. Now, we define $X={\left(x_{1},x_{2},x_{3},x_{4}\right)}^{T}$ such that Model (Equation 2) can be rewritten as $\frac{dX}{dt}=F\left(x\right)$, where $F=(f_{1},f_{2},f_{3},f_{4})$ Hence, Model (Equation 2) becomes:
(26)$$\left\{\begin{array}{c} \begin{array}{cc} f_{1} & =\frac{dx_{1}}{dt}=\Pi -\beta x_{1}x_{3}-\mu_{1}x_{1},\\ f_{2} & =\frac{dx_{2}}{dt}=\beta x_{1}x_{3}-\mu_{2}x_{2},\\ f_{3} & =\frac{dx_{3}}{dt}=px_{2}-\left(\mu_{3}+rx_{4}\right)x_{3},\\ f_{4} & =\frac{dx_{4}}{dt}=\frac{\alpha x_{2}}{x_{3}+\theta }-\mu_{4}x_{4}. \end{array} \end{array}\right.$$At $R_{0}=1$, we choose the bifurcation parameter $\tilde{\beta }$ such that
(27)$$\begin{array}{ccc} \tilde{\beta } & = & \beta^{*}=\frac{\mu_{1}\mu_{2}\mu_{3}}{\Pi p}. \end{array}$$Then, the Jacobian matrix of Equation (Equation 26) at the disease-free equilibrium $\bar{P}$ is given by
(28)$$\begin{array}{ccc} J_{\bar{P}} & = & \left[\begin{array}{cccc} -\mu_{1} & 0 & -\frac{\beta^{*}\;\pi }{\mu_{1}} & 0\\ 0 & -\mu_{2} & \frac{\beta^{*}\;\pi }{\mu_{1}} & 0\\ 0 & p & -\mu_{3} & 0\\ 0 & \frac{\alpha }{\theta } & 0 & -\mu_{4} \end{array}\right]. \end{array}$$To compute the right eigenvectors, $w={\left(w_{1},w_{2},w_{3},w_{4}\right)}^{T}$, we consider the system Jw=0
(29)$$\begin{array}{cc} -\mu_{1}w_{1}-\frac{\beta \;\pi \;w_{3}}{\mu_{1}}= & 0,\\ -\mu_{2}w_{2}+\frac{\beta \;\pi \;w_{3}}{\mu_{1}}= & 0,\\ pw_{2}-\mu_{3}w_{3}= & 0,\\ \frac{\alpha \;w_{2}}{\theta }-\mu_{4}w_{4}= & 0. \end{array}$$From Equation (Equation 29), we obtain
$$w_{1}=-\frac{\beta^{*}\Pi }{\mu_{1}^{2}}w_{3},w_{2}=\frac{\beta^{*}\Pi -\mu_{1}\mu_{3}}{\mu_{1}\left(p-\mu_{2}\right)}w_{3},w_{4}=\frac{\alpha }{\theta \mu_{4}}\left(\frac{\beta^{*}\Pi -\mu_{1}\mu_{3}}{\mu_{1}\left(p-\mu_{2}\right)}\right)w_{3}.$$Next, we compute the left eigenvector, $v=(v_{1},v_{2},v_{3},v_{4})$ from vJ=0 and the system becomes
(30)$$\begin{array}{cc} -\mu_{1}v_{1}= & 0,\\ -\mu_{2}v_{2}+pv_{3}+\frac{\alpha }{\theta }v_{4}= & 0,\\ -\frac{\beta^{*}\Pi }{\mu_{1}}v_{1}+\frac{\beta^{*}\Pi }{\mu_{1}}v_{2}-\mu_{3}v_{3}= & 0,\\ -\mu_{4}v_{4}= & 0. \end{array}$$From Equation (Equation 30), we obtain
$$v_{1}=v_{4}=0,v_{2}=\frac{\mu_{1}\left(\mu_{3}-p\right)}{\beta^{*}\Pi -\mu_{1}\mu_{2}}v_{3},$$
where $v_{3}$ is calculated to ensure that the eigenvectors satisfy the condition $v\dot{w}=1$. Since the first and fourth component of *v* are zero, we do not need the derivatives of $f_{1}$ and $f_{4}$. From the derivatives of $f_{2}$ and $f_{3}$, the only ones that are nonzero are the following:
$$\frac{\partial^{2}f_{1}}{\partial x_{1}\partial x_{3}}=\frac{\partial^{2}f_{1}}{\partial x_{3}\partial x_{1}}=-\beta^{*},\frac{\partial^{2}f_{2}}{\partial x_{1}\partial x_{3}}=\frac{\partial^{2}f_{2}}{\partial x_{3}\partial x_{1}}=\beta^{*}$$
$$\frac{\partial^{2}f_{3}}{\partial x_{3}\partial x_{4}}=\frac{\partial^{2}f_{3}}{\partial x_{4}\partial x_{3}}=-r,\frac{\partial^{2}f_{4}}{\partial x_{2}\partial x_{3}}=\frac{\partial^{2}f_{4}}{\partial x_{3}\partial x_{2}}=-\frac{\alpha }{\theta^{2}}$$
with
$$\frac{\partial^{2}f_{1}}{\partial x_{3}\partial \beta^{*}}=-\frac{\Pi }{\mu_{1}},\frac{\partial^{2}f_{2}}{\partial x_{3}\partial \beta^{*}}=\frac{\Pi }{\mu_{1}}$$All the other partial derivatives are zero. The direction of the bifurcation at $R_{0}=1$ is determined by the signs of the bifurcation coefficients a and b, obtained from the above partial derivatives, given respectively by:
$$\begin{array}{cc} a= & w_{1}v_{1}v_{3}\frac{\partial^{2}f_{1}}{\partial x_{1}\partial x_{3}}+w_{1}v_{3}v_{1}\frac{\partial^{2}f_{1}}{\partial x_{3}\partial x_{1}}+w_{2}v_{1}v_{3}\frac{\partial^{2}f_{2}}{\partial x_{1}\partial x_{3}}+w_{2}v_{3}v_{1}\frac{\partial^{2}f_{2}}{\partial x_{3}\partial x_{1}}\\ & +w_{3}v_{3}v_{4}\frac{\partial^{2}f_{3}}{\partial x_{3}\partial x_{4}}+w_{3}v_{4}v_{3}\frac{\partial^{2}f_{3}}{\partial x_{4}\partial x_{3}}+w_{4}v_{2}v_{3}\frac{\partial^{2}f_{4}}{\partial x_{2}\partial x_{3}}+w_{4}v_{3}v_{2}\frac{\partial^{2}f_{4}}{\partial x_{3}\partial x_{2}}\\ = & -w_{1}\left(v_{1}v_{3}+v_{3}v_{1}\right)\beta^{*}-w_{2}\left(v_{1}v_{3}+v_{3}v_{1}\right)\beta^{*}-w_{3}\left(v_{3}v_{4}+v_{4}v_{3}\right)r-w_{4}\left(v_{2}v_{3}+v_{3}v_{2}\right)\frac{\alpha }{\theta^{2}}\\ < & 0 \end{array}$$
and
$$\begin{array}{cc} b= & w_{1}v_{3}\frac{\partial^{2}f_{1}}{\partial x_{3}\partial \beta^{*}}+w_{2}v_{3}\frac{\partial^{2}f_{2}}{\partial x_{3}\partial \beta^{*}}\\ = & \frac{p\Pi }{\mu_{1}\mu_{2}}w_{2}v_{3}>0. \end{array}$$Therefore, Model (Equation 2) exhibits forward bifurcation at $R_{0}=1$. □

**Remark** **3.***In case of reinfection, the global stability of the endemic equilibrium $P^{*}$ is not guaranteed when $R_{0}>1$; this is due to some external factors. In the next subsection, we deal with the global stability of Model *(Equation 2)* at the endemic equilibrium point $P^{*}$ when $R_{0}>1$.*

Now, we study the stability of $P^{*}$.

**Theorem** **7.***Model *(Equation 2)* is locally asymptotically stable at $P^{*}$ if $R_{0}>1$; otherwise, it is unstable.*

**Proof.** At the endemic equilibrium $P^{*}$, the Jacobian matrix of Model (Equation 2) is given by:  □


(31)
$$\begin{array}{c} J\left(P^{*}\right)=\left[\begin{array}{cccc} -\beta \;V^{*}-\mu_{1} & 0 & -\beta \;E_{s^{*}} & 0\\ \beta \;V^{*} & -\mu_{2} & \beta \;E_{s^{*}} & 0\\ 0 & p & -A^{*}r-\mu_{3} & -rV^{*}\\ 0 & \frac{\alpha }{V^{*}+\theta } & -\frac{\alpha \;E_{i^{*}}}{{\left(V^{*}+\theta \right)}^{2}} & -\mu_{4} \end{array}\right] \end{array}$$


The characteristics in λ at the endemic equilibrium $P^{*}$ are given by
(32)$$\lambda^{4}+\sigma_{1}\lambda^{3}+\sigma_{2}\lambda^{2}+\sigma_{3}\lambda +\sigma_{4}=0.$$
where
$$\begin{array}{cc} \sigma_{1}= & rA+V\beta +\mu_{1}+\mu_{2}+\mu_{3}+\mu_{4},\\ \sigma_{2}= & (\beta \;\left(Ar+\mu_{2}+\mu_{3}+\mu_{4}\right)V^{3}\\ & +\left(2\;\beta \;\left(Ar+\mu_{2}+\mu_{3}+\mu_{4}\right)\theta +\left(Ar+\mu_{1}+\mu_{3}+\mu_{4}\right)\mu_{2}+\left(Ar+\mu_{1}+\mu_{3}\right)\mu_{4}-pE_{s}\beta +\mu_{1}\left(Ar+\mu_{3}\right)\right)V^{2}\\ & +\left(\beta \;\left(Ar+\mu_{2}+\mu_{3}+\mu_{4}\right)\theta^{2}+\left(\left(2\;Ar+2\;\mu_{1}+2\;\mu_{3}+2\;\mu_{4}\right)\mu_{2}+\left(2\;Ar+2\;\mu_{1}+2\;\mu_{3}\right)\mu_{4}-2\;pE_{s}\beta +2\;\mu_{1}\left(Ar+\mu_{3}\right)\right)\theta -E_{i}r\alpha \right)V\\ & +\left(\left(Ar+\mu_{1}+\mu_{3}+\mu_{4}\right)\mu_{2}+\left(Ar+\mu_{1}+\mu_{3}\right)\mu_{4}-pE_{s}\beta +\mu_{1}\left(Ar+\mu_{3}\right)\right)\theta^{2}),\\ \sigma_{3}= & \left(\left(Ar+\mu_{2}+\mu_{3}\right)\mu_{4}+\mu_{2}\left(Ar+\mu_{3}\right)\right)\beta \;V^{3}\\ & +\left(2\;\left(\left(Ar+\mu_{2}+\mu_{3}\right)\mu_{4}+\mu_{2}\left(Ar+\mu_{3}\right)\right)\beta \;\theta +\left(\left(Ar+\mu_{1}+\mu_{3}\right)\mu_{2}-pE_{s}\beta +\mu_{1}\left(Ar+\mu_{3}\right)\right)\mu_{4}+\mu_{1}\left(Ar+\mu_{3}\right)\mu_{2}-\left(\alpha \;\left(E_{i}-E_{s}\right)r+pE_{s}\mu_{1}\right)\beta \right)V^{2}\\ & +\left(\left(\left(Ar+\mu_{2}+\mu_{3}\right)\mu_{4}+\mu_{2}\left(Ar+\mu_{3}\right)\right)\beta \;\theta^{2}+\left(\left(\left(2\;Ar+2\;\mu_{1}+2\;\mu_{3}\right)\mu_{2}-2\;pE_{s}\beta +2\;\mu_{1}\left(Ar+\mu_{3}\right)\right)\mu_{4}+2\;\mu_{1}\left(Ar+\mu_{3}\right)\mu_{2}-2\;E_{s}\left(p\mu_{1}-1/2\;r\alpha \right)\beta \right)\theta -E_{i}r\alpha \;\left(\mu_{2}+\mu_{1}\right)\right)V\\ & +\left(\left(\left(Ar+\mu_{1}+\mu_{3}\right)\mu_{2}-pE_{s}\beta +\mu_{1}\left(Ar+\mu_{3}\right)\right)\mu_{4}+\left(\mu_{2}\left(Ar+\mu_{3}\right)-pE_{s}\beta \right)\mu_{1}\right)\theta^{2}\\ \sigma_{4}= & \mu_{2}\mu_{4}\beta \;\left(Ar+\mu_{3}\right)V^{3}\\ & +\left(\left(2\;\left(\beta \;\theta +1/2\;\mu_{1}\right)\left(Ar+\mu_{3}\right)\mu_{2}-p\beta \;E_{s}\mu_{1}\right)\mu_{4}-r\beta \;\alpha \;\left(E_{i}\mu_{2}-E_{s}\mu_{1}\right)\right)V^{2}\\ & +\left(\left(\left(\beta \;\theta +2\;\mu_{1}\right)\left(Ar+\mu_{3}\right)\mu_{2}-2\;p\beta \;E_{s}\mu_{1}\right)\theta \;\mu_{4}-r\alpha \;\mu_{1}\left(-\beta \;\theta \;E_{s}+E_{i}\mu_{2}\right)\right)V\\ & +\left(\mu_{2}\left(Ar+\mu_{3}\right)-pE_{s}\beta \right)\mu_{4}\mu_{1}\theta^{2} \end{array}$$

Clearly, $\sigma_{1}>0.$ Thus, using the Routh–Hurwitz criteria, we can say that the equilibrium $P^{*}$ of Model (Equation 2) is locally asymptotically stable if the following relations are true:(33)$$\begin{array}{c} \sigma_{2}>0,\;\sigma_{3}>0,\sigma_{4}>0,\;\sigma_{1}\sigma_{2}-\sigma_{3}>0 \end{array}$$(34)$$\begin{array}{c} \sigma_{1}\sigma_{2}\sigma_{3}-\sigma_{3}^{2}-\sigma_{4}\sigma_{1}^{2}>0. \end{array}$$

We now analyze the global stability of Model (Equation 2) for the endemic equilibrium point $P^{*}$ when $R_{0}>1$. To show this, we use a Dulac function. We prove the global stability of endemic equilibrium in the following theorem.

**Theorem** **8.***When $R_{0}>1$, Model *(Equation 2)* is globally asymptotically stable at the endemic equilibrium point $P^{*}$.*

The proof of the above theorem is given in Appendix B.

## 4. Dynamics of the System with Impulsive Drug Dosing

### 4.1. Dynamics of the Drug

To analyze the dynamics of the drug dosing, we first analyze the following sub-system.
(35)$$\begin{array}{ccc} \frac{dD}{dt} & = & -\mu_{5}D,\;\;\;\;t\neq t_{n}\\ \Delta D & = & \omega ,\;\;\;\;\;\;\;\;t=t_{n} \end{array}$$
where $\Delta =D\left(t_{n}^{+}\right)-D\left(t_{n}^{-}\right)$. Let $\tau =t_{k+1}-t_{n}$ be the period of the drug dosing. The solution of Equation (Equation 35) is
(36)$$\begin{array}{ccc} D(t) & = & D\left(t_{n}^{+}\right)e^{-\mu_{5}\left(t-t_{n}\right)},\;\;\;\mathrm{for}\;\;t_{n}<t\leq t_{k+1}. \end{array}$$

In the presence of impulsive dosing, we can obtain the recursion relation at the moments of impulse, as written below:$$\begin{array}{c} D\left(t_{n}^{+}\right)=D\left(t_{n}^{-}\right)+\omega . \end{array}$$

Thus, the concentrations of the drug before and after the impulse are obtained respectively as
(37)$$D\left(t_{n}^{+}\right)=\frac{\omega (1-e^{-k\tau \mu_{5}})}{1-e^{-\tau \mu_{5}}}$$
and
(38)$$D\left(t_{k+1}^{-}\right)=\frac{\omega \left(1-e^{-k\tau \mu_{5}}\right)e^{-\tau \mu_{5}}}{1-e^{-\tau \mu_{4}}}.$$

Thus, the limiting value of the drug concentration before and after one cycle are
$$\begin{array}{c} \lim_{k\to \infty }D\left(t_{n}^{+}\right)=\frac{\omega }{1-e^{-\tau \mu_{5}}}\;\;\mathrm{and}\;\;\lim_{k\to \infty }D\left(t_{k+1}^{-}\right)=\frac{\omega e^{-\tau \mu_{5}}}{1-e^{-\tau \mu_{5}}} \end{array}$$
and
$$\begin{array}{c} D\left(t_{k+1}^{+}\right)=\frac{\omega e^{-\tau \mu_{5}}}{1-e^{-\tau \mu_{5}}}+\omega =\frac{\omega }{1-e^{-\tau \mu_{5}}}, \end{array}$$
respectively.

We now require the following definitions and lemmas for this study [45,46]:

**Definition** **1.**
*Let $\Lambda \equiv (E_{S},E_{I},V,A,D)$, $B_{0}=\left[B:R_{+}^{5}\to R_{+}\right]$; then, we say that B belongs to class $B_{0}$ if the following conditions hold:*

*(i) B is continuous on $\left(t_{n},t_{k+1}\right]\times R_{+}^{5},n\in N$, and for all $\Lambda \in R^{5},$*

*${lim}_{\left(t,\mu \right)\to \left(t_{n}^{+},\Lambda \right)}$$B\left(t,\mu \right)=B\left(t_{n}^{+},\Lambda \right)$ exists;*
*(ii) B is locally Lipschitzian in* Λ.

**Lemma** **1.**
*Let Z(t) be a solution of the system (Equation 4) with $Z(0^{+})\geq 0$. Then, $Z_{i}\left(t\right)\geq 0$, i=1,…,5 for all t≥0. Moreover, $Z_{i}\left(t\right)>0$, i=1,…,5 for all t>0 if $Z_{i}\left(0^{+}\right)>0$, i=1,…,5.*


**Lemma** **2.**
*There exists a constant γ such that $E_{S}\left(t\right)\leq \gamma$, $E_{I}\left(t\right)\leq \gamma$, V(t)≤γ
D(t)≤γ for each and every solution Z(t) of Model (Equation 4) for all sufficiently large t.*


**Lemma** **3.**
*Let assume that $B\in B_{0}$ and also let*

$$\begin{array}{c} D^{+}B\left(t,Z\right)\leq j\left(t,B\left(t,Z\left(t\right)\right)\right),\;\;t\neq t_{n},\\ B\left(t,Z\left(t^{+}\right)\right)\leq \Phi_{n}\left(B\left(t,Z\left(t\right)\right)\right),\;\;t=t_{n}, \end{array}$$

*where $j:R_{+}\times R_{+}\to R$ is a continuous function in $\left(t_{n},t_{k+1}\right]$ for $e\in R_{+}^{2}$, n∈N, the limit $\lim_{\left(t,V\right)\to \left(t_{n}^{+}\right)}j\left(t,g\right)=j\left(t_{n}^{+},x\right)$ exists, and $\Phi_{n}^{i}\left(i=1,2\right):R_{+}\to R_{+}$ is non-decreasing. Let y(t) be a maximal solution of the following impulsive differential equation:*

(39)
$$\begin{array}{ccc} \frac{dx(t)}{dt} & = & j\left(t,x\left(t\right)\right),\;t\neq t_{n},\\ x(t^{+}) & = & \Phi_{n}\left(x\left(t\right)\right),\;t=t_{n},\;x\left(0^{+}\right)=x_{0}, \end{array}$$

*existing on $\left(0^{+},\infty \right)$. Then, $B\left(0^{+},Z_{0}\right)\leq x_{0}$ implies that B(t,Z(t))≤y(t),t≥0 for any solution Z(t) of Model (Equation 4). If j satisfies additional smoothness conditions to ensure the existence and uniqueness of solutions for (Equation 39), then y(t) is the unique solution of (Equation 39).*


The lemmas provided above give the following result:

**Lemma** **4.**
*Model (Equation 35) has a unique positive periodic solution $\tilde{D}\left(t\right)$ with period τ and can be written as*

(40)
$$\tilde{D}\left(t\right)=\frac{\omega \;exp(-\mu_{5}\left(t-t_{n}\right))}{1-exp(-\tau \mu_{5})},\;t_{n}\leq t\leq t_{k+1},\;\tilde{D}\left(0^{+}\right)=\frac{\mu_{5}}{1-exp(-\tau \mu_{5})}.$$



In the above section, we discussed perfect drug dosing. Now, we discuss imperfect drug dosing in the following subsection.

#### 4.1.1. Impact of Imperfect Drug Dosing

Suppose a COVID-19 patient during the treatment stage takes a drug holiday after taking $n=n_{1}$ doses. Let us assume a positive quantity $D_{1}$, which denotes the minimum difference between the concentration of the drug after $n=n_{1}$ doses and the normal concentration of drug $D_{2}$.

Our consideration is based on the assumption that a patient usually misses his perfect drug dose when he achieves the almost cured stage. Thus, COVID-19 patients can take a drug holiday after taking $n_{1}$ doses when the difference between the drug concentrations after $n=n_{1}$ doses and normal threshold $\tilde{A}$ is less than a chosen small positive number, $D_{1}$. Thus, we must have
(41)$$\begin{array}{ccc} D(t_{n_{1}}^{+}) & \geq & D_{2}-D_{1}. \end{array}$$

A patient may take a drug holiday once the drug concentration reaches a periodic orbit. Suppose $h_{1}$ doses are subsequently missed; then, we impose the condition that the drug concentration reach a high level and the patient can realize that the further treatment is highly needed.

Now, we assume that the difference between the present drug concentration and its possible maximum response is less or equal to a small positive number (ϵ). Therefore, the inequality $D\left(t_{n_{1}+h_{1}}^{-}\right)\geq D_{max}-\epsilon$ allows us to find the maximum number of doses a patient can miss.

Suppose a patient has missed $h_{1}$ doses. Now, in order to keep the drug concentration above $D_{2}$ after $n_{2}$ doses are taken, a new condition must be applied that forces the drug concentration level to $D_{3}$ away from the $D_{2}$. The condition is that the following that must be satisfied:(42)$$\begin{array}{ccc} D(t_{n_{1}+h_{1}+n_{2}}^{+}) & \geq & D_{2}-D_{3}. \end{array}$$

From (Equation 47), $n_{2}$ can be determined. However, the calculations for determining $n_{2}$ are complicated. One can see [47] for detailed analysis. We determine $n_{2}$ from numerical simulations.

### 4.2. Dynamics of the Impulsive System (Equation 4)

Using the result in Lemma 4, we establish the following theorem.

**Theorem** **9.**
*The disease-free periodic orbit $\left(\tilde{E_{S}},0,0,\tilde{A},\tilde{D}\right)$ of Model (Equation 2) is locally asymptotically stable if*

(43)
$$\begin{array}{c} \tilde{R_{0}}<1 \end{array}$$

*where*

$$\begin{array}{c} \tilde{R_{0}}=\frac{\mu_{2}}{p\beta \;\tau }\int_{0}^{\tau }\frac{\left(\mu_{3}+r\tilde{A}\right)}{{\tilde{E}}_{S}}dt. \end{array}$$



**Proof.** Let the disease-free periodic solution of Model (Equation 4) be denoted by $\tilde{P}\left(\tilde{E_{S}},0,0,\tilde{A},\tilde{D}\right)$, where
$$\tilde{D}\left(t\right)=\frac{\omega \;\exp(-\mu_{5}\left(t-t_{n}\right))}{1-\exp(-\tau \mu_{5})},\;t_{n}\leq t\leq t_{k+1},$$
with the initial condition $D(0^{+})$ as in Lemma 4.We now test the stability of the disease-free equilibrium point. The variational matrix M(t) at the disease-free periodic orbit $\tilde{P}\left(\tilde{E_{S}},0,0,\tilde{A},\tilde{D}\right)$ is calculated as
$$M\left(t\right)=\left[m_{ij}\right]=\left(\begin{array}{ccccc} -\mu_{1} & 0 & 0 & \;\;-\beta {\tilde{E}}_{S} & \;\;\;0\\ \;\\ 0 & -\mu_{2} & \beta {\tilde{E}}_{S} & \;\;0 & \;\;\;0\\ \;\\ 0 & p & -\mu_{3}-r\tilde{A} & \;\;0 & \;\;\;0\\ \;\\ 0 & \frac{\alpha }{\theta } & 0 & \;\;-\mu_{4} & \;\;\;\zeta \\ \;\\ 0 & \;0 & \;0 & \;0\;\; & \;\;\;-\mu_{5} \end{array}\right).$$The monodromy matrix P of the variational matrix M(t) is
$$P\left(\tau \right)=I_{n}\exp\left(\int_{0}^{\tau }M(t)dt\right),$$
where $I_{n}$ is the identity matrix.The monodromy matrix can be rewritten as $P\left(\tau \right)=\mathrm{diag}\left(\sigma_{1},\sigma_{2},\sigma_{3},\sigma_{4}\right)$, where $\sigma_{i},\;i=1,2,3,4$, are the Floquet multipliers, determined as
$$\begin{array}{ccc} & & \sigma_{1}=\exp\left(-\mu_{1}\tau \right),\;\;\sigma_{2,3}=\exp\left(\int_{0}^{\tau }\frac{1}{2}\left[-A\pm \sqrt{a^{2}-4b}\right]dt\right),\\ & & \sigma_{4}=\exp\left(-\mu_{4}\tau \right),\;\;\;\sigma_{5}=\exp\left(-\mu_{5}\tau \right). \end{array}$$Here, $a=\mu_{2}+\mu_{3}+\tilde{A}$ and $b=\mu_{2}\left(\mu_{3}+\tilde{A}\right)-p\beta {\tilde{E}}_{S}$. It is noted that $\sigma_{1,4,5}<1$. Furthermore, we check that $a^{2}-4b>0$. Now, if b≥0 holds, then we obtain $\sigma_{2,3}<1$. Thus, according to Floquet theory, the disease-free periodic orbit $\tilde{P}\left(\tilde{E_{S}},0,0,\tilde{A},\tilde{D}\right)$ of Model (Equation 4) is asymptotically stable if the conditions given in (Equation 43) are true. □

There exists another periodic orbit $P({\tilde{E}}_{S},{\tilde{E}}_{I},\tilde{V},\tilde{A},\tilde{D})$ of impulsive Model (Equation 4). We analyze its dynamics through numerical simulation.

## 5. Numerical Simulation

In this section, we study the mathematical models (Equation 2) through numerical simulation. The values of the model parameters used in the numerical simulations are taken from Table 1; some are varied for to study different dynamical regimes.

For the dynamical simulation of the model without drugs (i.e, Model (Equation 2)), we take the initial conditions as: $H\left(0\right)=4\times 10^{5}$ cells mL${}^{-1}$, $I\left(0\right)=5\times 10^{-4}$ cells mL${}^{-1}$, V(0)=300 RNA copies mL${}^{-1}$, and A=0 molecules mL${}^{-1}$.

Figure 2 describes the forward bifurcation of Model (Equation 2) using Theorem 6 and the parameters value in Table 1. When $R_{0}>1$, the endemic equilibrium exists. The disease-free state loses its stability when $R_{0}>1$.

The disease-free equilibrium is stable when $R_{0}<1$, while the endemic equilibrium starts to rise with $R_{0}>1$ (Theorem 4). Figure 3 represents a region of stability of the equilibrium points in parametric planes. In Figure 3a, as both β and *p* increase from lower to higher values, the disease equilibrium becomes unstable in the region where $R_{0}>1$, and the endemic state becomes feasible. In Figure 3b, we observe that as the clarence rate of the viral load is increased, the area of stability of the disease-free state is increased. This can be accomplished using drug dosing.

Figure 4 shows the solution trajectories with time. For the set of values used, $R_{0}$ is greater than one. That means the system is in an endemic state. Now, as the infection rate is increased, susceptible cells decrease and both the infected cell population and the viral load increase accordingly (Figure 5).

Figure 6 shows that the phase trajectories converge to the same point (endemic equilibrium point), though the initial values are different. From this, we can conclude that the endemic equilibrium is globally asymptotically stable.

The numerical solution of Model (Equation 4) is plotted in Figure 7. The virus population is increasing, whereas susceptible cell density decreases due to the infection. Without drug application, the antibody response is low (Figure 7b) and, thus, infected cells or viruses are increased.

Numerical solutions of the impulsive system for dosing rates (ω=100 mg) are plotted in Figure 7 for a fixed impulse interval τ=7 day. We may conclude that for quick recovery, higher doses should be taken. Figure 8 plots two different intervals of impulses. We can see that the lower interval (τ=7 day) is capable of achieving disease-free periodic orbit sooner than the higher interval, in τ=14 days. Additionally, we found that with a higher interval of impulse (τ=14 days) with higher dosing (200 mg), the system does not converge to the
disease-free periodic orbit (Figure 8).

In Figure 9, we can observe the dynamics of the drug for imperfect drug dosing corroborated with the Section 4.1.1. In order to show the impact of taking drug holidays, we find the time at which a SARS-CoV-2 patient can take the required number of doses and then miss the maximum number of doses. From this figure, we obtain the number of possible maximum drug holidays during the treatment period for a fixed drug dose and dosing interval. Taking the drug dose ω=100 mg and a dosing interval τ=7 day, the maximum number of holidays is fourteen days; i.e., two doses can be missed after 9 doses ($n_{1}=9$). This figure also shows that after two consecutive drug holidays ($h_{1}=2$), if the patient again takes five doses (i.e., $n_{2}=5$), then the antibody response will gain its previous equilibrium position (periodic orbit).

## 6. Discussion and Conclusions

In this study, we used classical susceptible or uninfected cells, infected cells, and virus population in the presence of adaptive immune responses as a functional form. More attention is given to antibody-response modeling and the role of immune responses against the invading SARS-CoV-2 virus in our respiratory system, which is the primary target area.

We computed the basic reproduction number $R_{0}$ for our model. We observed that the model system has two equilibrium points; one is disease-free equilibrium ($\bar{P}$) and the other is endemic equilibrium ($P^{*}$). The disease-free equilibrium is stable asymptotically when the basic reproduction number $R_{0}$ is below the unity. When $R_{0}$ is greater than unity, the disease-free equilibrium becomes unstable and endemic equilibrium becomes feasible. Here, $R_{0}=1$ is the forward transcritical bifurcation point at which the system switches its stability from disease-free to endemic equilibrium.

Finally, we studied the effects of taking the drug in impulsive mode and with holidays during treatment. The numerical simulation of an impulse dosing interval of τ=7 days and ω=100 mg dosing rate can achieve a disease-free state in a short time. This study also shows that for a short treatment period, instead of taking the drug at every one-day interval for the entire length of the induction period, it would be better if the patient takes two one-day drug holidays after taking the first twenty-two doses.

In a nutshell, the proposed impulsive mathematical model is functional. It successfully describes the dynamics of SARS-CoV-2 within humans. The results obtained from this study can further guide development of a cost-effective drug regimen for SARS-CoV-2 patients with fewer side effects.

## Figures and Tables

**Figure 1 vaccines-10-01846-f001:**
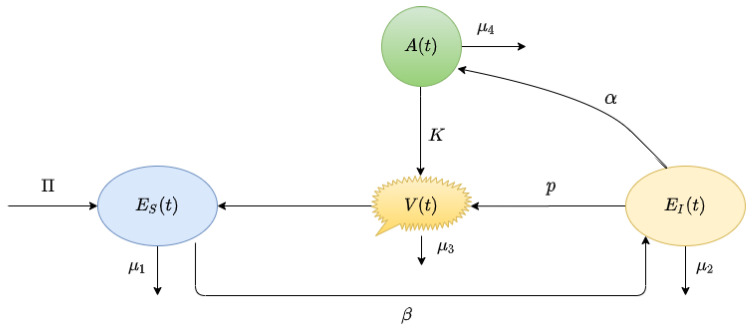
Conceptual diagram of Model (Equation 2). It shows the flowchart of antibody responses in SARS-CoV-2 infection within a host.

**Figure 2 vaccines-10-01846-f002:**
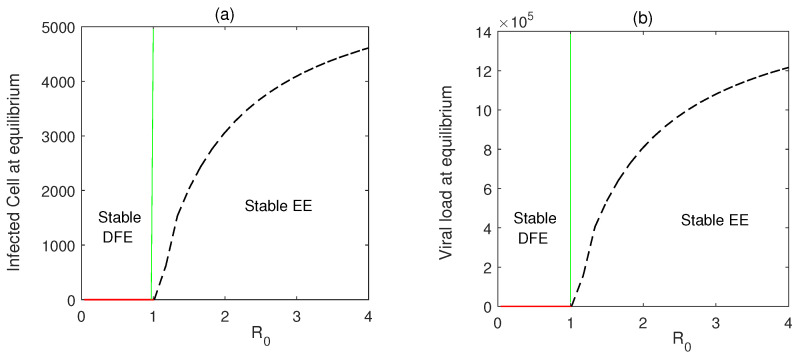
(**a**,**b**) Forward bifurcation diagram of Model (Equation 2) using Theorem 6. Red curves represent stable disease-free equilibria (DFE) and the black-dashed line denotes stable endemic equilibria (EE).

**Figure 3 vaccines-10-01846-f003:**
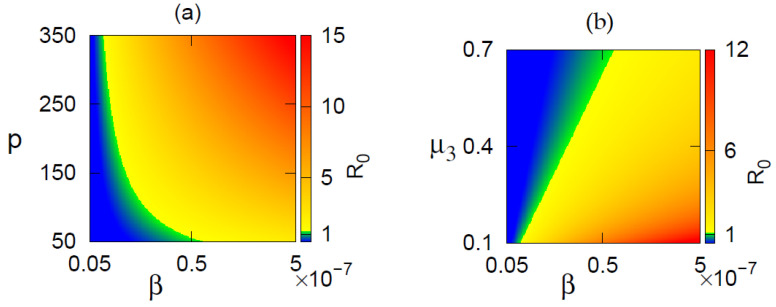
Region of stability of disease-free equilibrium (DFE) and endemic equilibrium (EE) shown in (**a**) β−p, (**b**) $\beta -\mu_{3}$ parameter planes. Color represents the value of $R_{0}$. DFE is stable for $R_{0}<1$ and unstable for $R_{0}>1$. EE exists and is stable for $R_{0}>1$.

**Figure 4 vaccines-10-01846-f004:**
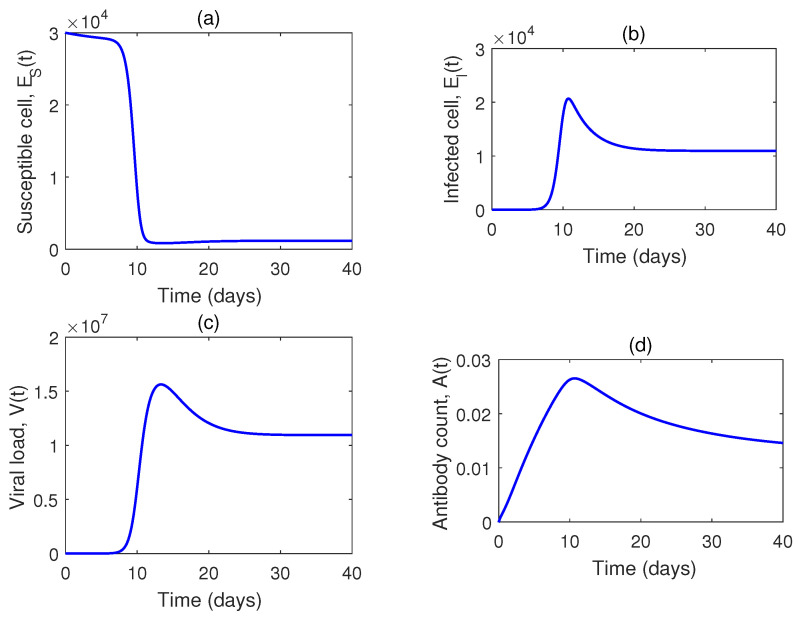
(**a**–**d**) Numerical solution of Model (Equation 2) for the set of parameters in Table 1.

**Figure 5 vaccines-10-01846-f005:**
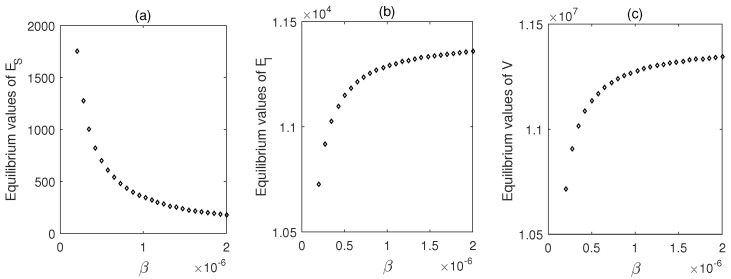
(**a**–**c**) Steady-state values of the populations plotted as function of β. Parameter values are same as in Figure 4.

**Figure 6 vaccines-10-01846-f006:**
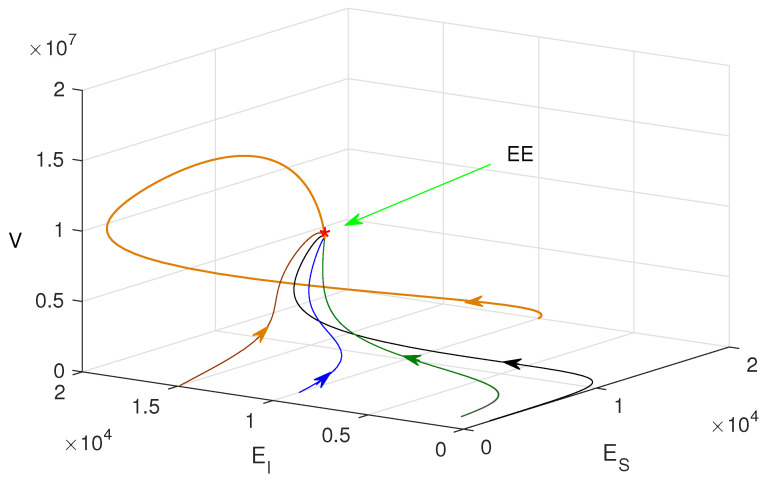
Phase portraits plotted in $E_{S}-E_{I}-V$ phase-space with different initial conditions and $R_{0}>1$.

**Figure 7 vaccines-10-01846-f007:**
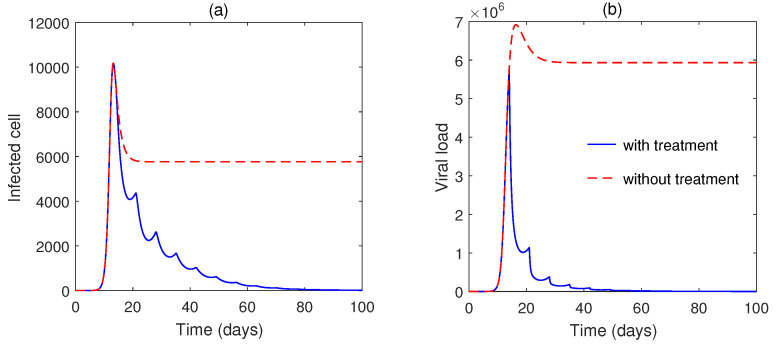
(**a**,**b**) Numerical solutions of impulsive Model Equation 4 with treatment (ω=60,τ=7) and without treatment.

**Figure 8 vaccines-10-01846-f008:**
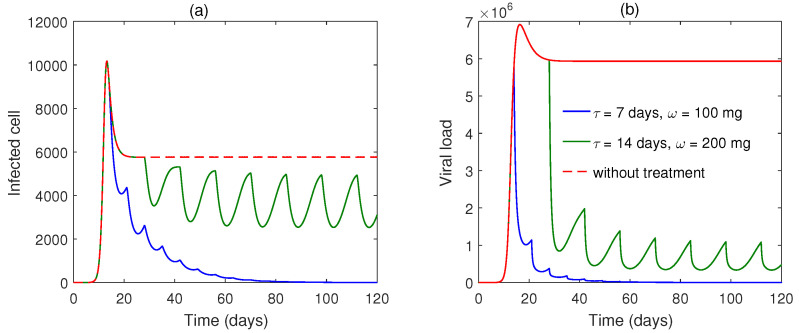
(**a**,**b**) Solutions of impulsive Model (Equation 4) shown for impulse interval τ=7 days with dosing rate ω=100 mg (blue line), and impulse interval of τ=14 days with dosing rate ω=200 mg (green line).

**Figure 9 vaccines-10-01846-f009:**
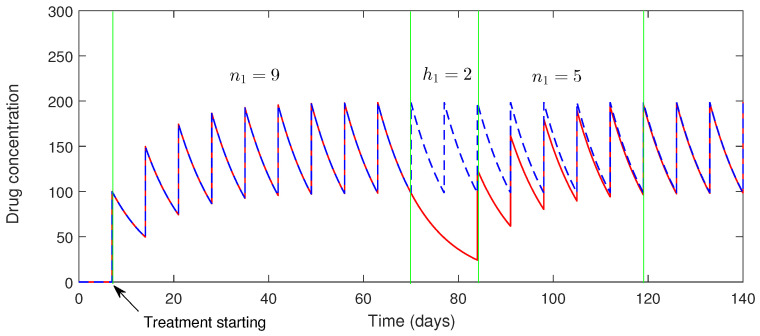
Dynamics of the drug with and without drug holidays, taking an impulse interval of (τ=7 days) and a fixed drug dosing of ω=100 mg with two consecutive drug holidays, $h_{1}=2$.

**Table 1 vaccines-10-01846-t001:** Short descriptions and values of the parameters of Models (Equation 2) and (Equation 4).

Parameters	Short Description	Value (Unit)	References
Π	Growth rate of epithelial cells	$4\times 10^{3}$ cells mL${}^{-1}$ day${}^{-1}$	[48]
$\mu_{1}$	Natural death rate of uninfected epithelial cells	0.2 day${}^{-1}$	[11,48]
$\mu_{2}$	Blanket death rate of infected epithelial cells	0.65 day${}^{-1}$	[11]
β	Rate of infection	(5–561) $\times 10^{-9}$ mL (RNA copies)${}^{-1}$ day${}^{-1}$	[24,31]
*p*	Growth rate of virus in cells	8.2–525 day${}^{-1}$	[48]
$\mu_{3}$	Virus clearance rate	(0–1) day${}^{-1}$	[24]
α	Rate of antibody response from immune cells	0–1 day${}^{-1}$	[48]
*r*	Viral particles’ rate of neutralization by antibodies	0–1 mL (molecules)${}^{-1}$ day${}^{-1}$	[48]
θ	Half-maximal simulation threshold	0.5 (RNA copies) mL${}^{-1}$	[48]
$\mu_{4}$	Antibody clearance rate	0.07 day${}^{-1}$	[48]
ζ	Antibody production rate by drug	6 molecules day${}^{-1}$ gm${}^{-1}$	Assumed
$\mu_{5}$	Decay rate of drug	0.1 mg day${}^{-1}$	Assumed

## Data Availability

The data used for supporting the findings are included within the article.

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
