# Peer review of "Global Dynamics of SARS-CoV-2 Infection with Antibody Response and the Impact of Impulsive Drug Therapy"

_vaccines, 2022, doi:10.3390/vaccines10111846_

Round 1

Reviewer 1 Report

Authors reported that global dynamics of SARS-CoV-2 infection with antibody response and the impact of impulsive drug therapy.

Which is this manuscript review or article?

Article should contain Methods and Result section.

Article may need statistical analysis.

There are many mathematical expressions.

Most readers may be difficult for many mathematical expressions.

This topic may not be suitable for this journal.

Take for message or the clinical usefulness may be important in this journal.  

Author Response

  1. Which is this manuscript review or article?

The manuscript is an article (research article).

  1. Article should contain Methods and Result section.

The article is based on mathematical modelling of the disease and it is designed as per standard format. This is a special issue paper.

  1. Article may need statistical analysis.

This article is a theoritical work and the analytical findings are verified with the statistical data in numericalsection.

  1. There are many mathematical expressions.

This work is based on dynamicall model formulation of COVID-19 disease and its analytical study. Thus many mathematical expression is quite relevent.

  1. Most readers may be difficult for many mathematical expressions.

Yes it is true, but the bilogical interpretation from analytical findig wills enreach the readers.

Reviewer 2 Report

The authors describe the kinetics of the SARS-CoV-2 virus using a mathematical model.  Their paper is very interesting and represents an alternative method to deal with the kinetics of an epidemic, moreover a pandemic.  The authors need to correct the use of the English language.  They also need to refer to the limitations of their model.  However, the study is very interesting and the mathematical models novel.

Author Response

Thank you very much for your comments. We have revised the manuscript.

Reviewer 3 Report

The manuscript describes a mathematical model of in-host viral infection that includes an antibody response and periodic drug treatment. There are multiple issues with the model and methodology (outlined below), but I also don't see how this manuscript fits within the scope of the journal. This journal seems to be primarily for clinical/experimental work (this paper has none of that) about vaccines. The model presented here does not examine a vaccine. Specific issues with the manuscript:

1. The paper is full of incorrect terminology, making it hard to follow what the authors are doing. Some examples (not exhaustive): "repertory track" instead of "respiratory tract" in the abstract; "People are assumed to be most infectious
when they are most indicative." (not sure what indicative means here); MERS-CoV is denoted as MARS-CoV. They refer to their model as "target-cell-limited", meaning that the infection is self-limiting due to lack of target cells, but since they include target cell generation, this model is NOT target-cell-limited.

2. The references are sometimes not relevant to the corresponding statement. The reference for the statement "People are assumed to be most infectious
when they are most indicative." is a manuscript about SARS-CoV-2 spread in epithelial cultures. The reference for the statement "This response is of great importance in preventing new infections during the activation of the adaptive immune system" is a modeling study of Leishmaniasis. The reference for the statement "Proteins of the natural immune system in a healthy cell also respond against the invading pathogens within the first minutes or hours of infection" is a paper describing the immune response to non-enveloped viruses (SARS-CoV-2 is enveloped). Reference 2 is an arXiv pre-print from 2020 --- surely there are similar modeling studies that have managed to survive peer review and get published in 2 years.  

3. There is also a lack of references for much of the introductory material, particularly for statements about the biological processes they are ostensibly modeling (see the 2nd paragraph on page 3).

4. The introduction is very muddled. There is a lot of discussion (and self-citation) of epidemiological models that are not really relevant for the current in-host study. There is not a good review of previously published with-in host models of SARS-CoV-2 (some of which include immune responses). The description of SARS-CoV-2 immune responses is also muddled and not focused on the aspects of the immune response they are trying to replicate.

5. The authors are not clear about which immune response they are modeling --- it is mostly referred to as antibodies, but sometimes as CTLs.

6. I'm curious as to why removal of antibodies when they bind to virus is not included in the model.

7. The parameters presented in Table 1 do not have units, so there is no way to tell if these are reasonable for SARS-CoV-2. While there is a citation here, the parameter values presented here do not match the values in the cited paper. Also the cited paper does not have a model with antibodies, so I don't know where the antibody parameter numbers came from.

8. The authors give no justification for the impulsive function on antibodies. What drug has this kind of effect? Monoclonal antibody treatment is a one-time infusion.

9. The authors assume a quasi-steady state approximation for antibodies to do their analysis. Is this really valid? The impulse function in antibodies causes a rapid variation in antibody levels and there is the slower time scale increase over the early part of the infection, so when are antibodies actually in a quasi-steady state?

10. In the analysis of the impulsive model, the authors assume only exponential decay of the antibodies between impulses; what happened to the growth term?

11. Line 371 states "The data are taken mainly from Wolfel et al. [36], and some are estimated." There is no patient data in this manuscript. (I believe this is the data used in the paper cited in Table 1, but as noted above, the parameters here don't match those of reference 16).

12. Figures 2: These figures make no sense. If these are not on a log scale, are you really starting with 5 infected cells and reaching an endemic state with 2 infected cells? If it is on a log scale, 10^5 initially infected cells seems ridiculously high. Also, I would expect 10^5 infected cells to produce way more than 600ish virions. There's similar scale issues with all the figures showing a time course.

13. Figure 4, should it be "Unstable for DFE" rather than "Unstable for EE"? The right hand side is currently labelled both stable and unstable for EE.

14. Figure 5 is labelled as showing variables over time, but there is no time in this figure.

Author Response

The manuscript describes a mathematical model of in-host viral infection that includes an antibody response and periodic drug treatment. There are multiple issues with the model and methodology (outlined below), but I also don't see how this manuscript fits within the scope of the journal. This journal seems to be primarily for clinical/experimental work (this paper has none of that) about vaccines. The model presented here does not examine a vaccine. Specific issues with the manuscript:
1. The paper is full of incorrect terminology, making it hard to follow what the authors are doing. Some examples (not exhaustive): "repertory track" instead of "respiratory tract" in the abstract; "People are assumed to be most infectious when they are most indicative." (not sure what indicative means here); MERS-CoV is denoted as MARS-CoV. They refer to their model as "target-cell-limited", meaning that the infection is self-limiting due to lack of target cells, but since they include target cell generation, this model is NOT target-cell-limited. - We have rechecked and corrected the issues raised by the reviewer.
2. The references are sometimes not relevant to the corresponding statement. The reference for the statement "People are assumed to be most infectious when they are most indicative." is a manuscript about SARS-CoV-2 spread in epithelial cultures. The reference for the statement "This response is of great importance in preventing new infections during the activation of the adaptive immune system" is a modeling study of Leishmaniasis. The reference for the statement "Proteins of the natural immune system in a healthy cell also respond against the invading pathogens within the first minutes or hours of infection" is a paper describing the immune response to non-enveloped viruses (SARS-CoV-2 is enveloped). Reference 2 is an arXiv pre-print from 2020 --- surely there are similar modeling studies that have managed to survive peer review and get published in 2 years. - We have rechecked and corrected the issues raised by the reviewer.
3. There is also a lack of references for much of the introductory material, particularly for statements about the biological processes they are ostensibly modeling (see the 2nd paragraph on page 3). - We have rechecked and corrected the issues raised by the reviewer.
4. The introduction is very muddled. There is a lot of discussion (and self-citation) of epidemiological models that are not really relevant for the current in-host study. There is not a good review of previously published with-in host models of SARS-CoV-2 (some of which include immune responses). The description of SARS-CoV-2 immune responses is also muddled and not focused on the aspects of the immune response they are trying to replicate. - We have rechecked and corrected the issues raised by the reviewer.
5. The authors are not clear about which immune response they are modeling --- it is mostly referred to as antibodies, but sometimes as CTLs. - We have rechecked and corrected the issues raised by the reviewer. The term CTL is changed to antibody. .
6. I'm curious as to why removal of antibodies when they bind to virus is not included in the model. Every model has certain limitation. In this model our main focus is to study the effect of SARS-CoV-2 specific antibody under impulsive control.
7. The parameters presented in Table 1 do not have units, so there is no way to tell if these are reasonable for SARS-CoV-2. While there is a citation here, the parameter values presented here do not match the values in the cited paper. Also the cited paper does not have a model with antibodies, so I don't know where the antibody parameter numbers came from. The Units are included in the revised version
8. The authors give no justification for the impulsive function on antibodies. What drug has this kind of effect? Monoclonal antibody treatment is a one-time infusion. Every antiviral drug therapy has a time lag between dosages. During this time, the using impulse differential equation model can study the drug concentration and its effect on the system. Thus, we introduce the impulsive differential model to find the optimal drug dosing interval and amount of drug dosage.
9. The authors assume a quasi-steady state approximation for antibodies to do their analysis. Is this really valid? The impulse function in antibodies causes a rapid variation in antibody levels. There is a slower time scale increase over the early part of the infection, so when are antibodies actually in a quasi-steady state? - We confirm that it is valid. We have also provided a proper reference for these calculations.
10. In the analysis of the impulsive model, the authors assume only exponential decay of the antibodies between impulses; what happened to the growth term? - We have analyzed the linealinearizedem of our impulsive system which is nonlinear. A nonlinear impulsive system is complex to anato analytically.
11. Line 371 states "The data are taken mainly from Wolfel et al. [36], and some are estimated." There is no patient data in this manuscript. (I believe this is the data used in the paper cited in Table 1, but as noted above, the parameters here don't match those of reference 16). - We have clarified this issue in the revised manuscript.
12. Figures 2: These figures make no sense. If these are not on a log scale, are you starting with 5 infected cells and reaching an endemic state with 2 infected cells? If it is on a log scale, 10^5 initially infected cells seemseemiculously high. Also, I would 10^5 infected cells to produce way mohan 600ish virions.There ares similar scale issues with all the figures showing a time course. -This issue has been solved in the revised manuscript.
13. Figure 4, should it be "Unstable for DFE" rather than "Unstable for EE"? The right-hand side is currently labeled both stable and unstable for EE. - We have checked and corrected the figures.
14. Figure 5 is labeled as showing variables over time, but there is no time in this figure. - We have checked and corrected the figures.

Round 2

Reviewer 1 Report

#1 Ro should be explained.

#2 Last sentence of abstract should be rewritten. "Take home message" may be helpful.  

#3 Text is difficult for many readers.

Author Response

Response to the Reviewer 1:

Comment #1: Ro should be explained.

Response:  has now been explained in the revised manuscript (see Remark 1, page no. 8 and Discussion section).

Comment #2: Last sentence of abstract should be rewritten. "Take home message" may be helpful.

Response:  Last sentence of abstract has been rewritten.

Comment #3: Text is difficult for many readers.

Response: We have checked and revised the English of the manuscript. Hope revised version is okay for the reader.

Reviewer 2 Report

The authors have made the necessary corrections. The manuscript can now be accepted for publication.

Author Response

Thank you for your comment.

Reviewer 3 Report

The majority of my previous comments have not been addressed --- stating that it has been checked without explaining what exactly has been changed is not sufficient. So here is the list again with notes on what has/has not been completed:

1. The paper is full of incorrect terminology, making it hard to follow what the authors are doing. My list was not exhaustive, but the authors appeared to not have bothered to check for any incorrect terminology themselves. They also have not corrected two of the mistakes I pointed out --- 'MERS' is still written as 'MARS' and the authors refer to their model as target-cell limited (it is not).
2. The references are sometimes not relevant to the corresponding statement. The specific papers I listed previously have been addressed, but there are still several references are not relevant to the corresponding statement.
3. There is also a lack of references for much of the introductory material. There isn't a single reference in the first paragraph. There are also many "statements of fact" in lines 76-108 that still don't have references.
4. The introduction is very muddled. The discussion of epidemiological models has not been removed. It is absolutely not relevant here and is simply there for self-citations. On the other hand, I still don't see a great discussion of SARS-CoV-2 in-host models. There have been a number of these published recently, including one that models monoclonal antibody treatment. Since the authors are also trying to model antibodies, it seems like that should be relevant.
5. The authors are not clear about which immune response they are modeling. This has been fixed.
6. I'm curious as to why removal of antibodies when they bind to virus is not included in the model. Saying that every model has limitations is a cop-out. Since you are trying to model antibodies, and presumably model them as realistically as possible, why is antibody loss to binding not considered? Have you tried it and it doesn't have a large effect? Do you not want to include it because it might make the drug look less effective?
7. The parameters presented in Table 1 do not have units. Units have been added to the table, although they are not all correct. Theta should have units of virus. alpha should have units of [virus]/[time][cell]. I'm also curious as to why uninfected cells are dying faster than infected cells (0.2/d for uninfected cells, only 0.189/d for infected cells). Being infected doesn't usually make cells live longer.
8. The authors give no justification for the impulsive function on antibodies. I understand that antivirals are taken as multiple doses. I would like the authors to specifically list which antiviral causes an increase in antibodies. The only treatment I know of that directly increases antibodies is monoclonal antibody treatment and that is a single infusion, so no time lag. The manuscript has been submitted to an experimental/clinical journal; the authors should be clear about the actual clinical application of their work.
9. The authors assume a quasi-steady state approximation for antibodies to do their analysis. Is this really valid? I don't see any reference listed near where the authors make the quasi-steady state assumption. As I mentioned previously, the quasi-steady-state approximation is NOT valid over the rapidly-varying time frame of individual doses. It is also NOT valid over the 20-30 day phase of rising antibodies. So where is it valid? The authors should be able to justify this with actual calculations of the time scales.
10. In the analysis of the impulsive model, the authors assume only exponential decay of the antibodies between impulses; what happened to the growth term? First of all, it is not clear that the system was "linearized." Second, linearizing a system does not mean ignoring all the nonlinear terms; it means using the first order (linear approximation) of the nonlinear terms. So this is not done correctly.
11. Line 371 states "The data are taken mainly from Wolfel et al. [36], and some are estimated." This sentence remains completely unchanged and I don't see any "clarification" that makes it accurate (there is NO PATIENT DATA in this paper).
12. Figures 2: These figures make no sense. I'm not sure how this was addressed because these figures are unchanged. 
13. Figure 4, should it be "Unstable for DFE" rather than "Unstable for EE"? This figure is also unchanged and therefore still incorrect.
14. Figure 5 is labeled as showing variables over time, but there is no time in this figure. The authors have not checked and corrected the figures because this one is also unchanged.

Round 3

Reviewer 1 Report

Revised manuscript has been improved.

Author Response

Thank you for your suggestions and comments.

Reviewer 3 Report

The authors have made a more serious attempt to address my comments this time around, but many issues still remain:

1. The authors are still referring to this model as 'target-cell limited.' In fact, they have added at least one additional reference to the model as target-cell limited. A target-cell limited model is one where the viral load goes to zero because there are no target cells left to infect. As their own simulations show, the viral load does not go to zero and susceptible cells are not depleted, so this is NOT a target-cell limited model.

2. The introduction is somewhat improved, but I would hesitate to say that there is sufficient discussion of previous SARS-CoV-2 models that incorporate antibodies.

3. While the authors say that they have checked that removal of antibodies via binding is not a significant effect, they have not included any evidence of this either in their response or in the manuscript. 

4. The authors have cleaned up the units in Table 1, but they also changed the values of many of the parameters, some of them by several orders of magnitude. One of the papers they now cite as a reference for the parameters is a model of dengue. How are they sure that they are actually modeling SARS-CoV-2? In fact, their simulations seem to suggest that these parameters are not realistically reproducing SARS-CoV-2. Fig. 4 shows viral load peaking at day 30 or later --- most clinical data suggests that SARS-CoV-2 viral load peaks about 5-10 days post-infection. The simulations also show chronic viral infections. While COVID patients sometimes have long-lasting symptoms, this is not associated with persistent viral infection.

5. There was no response to point 8 of the last round. Exactly what drug is being modeled here?

6. In response to my comment about ignoring antibody growth during analysis of the impulsive model, the authors point to one of their previous publications. The model in that paper is not applying an impulsive function to antibodies --- it is simply the addition of some external drug. In this case, you are adding to a component that has its own internal dynamics between the impulses. It's not clear that this can be ignored and the authors have not shown any evidence that it is appropriate to ignore it. 

7. The caption of Figure 2 does not match the figures.

8. There are several unknown references (??) in the manuscript.

9. I believe the figures showing the impulsive treatment were not re-done using the new parameter values. There is a huge difference in the scale of infected cells and virus between figures 8/9 and figure 4.
